# The Characterization of Stress Corrosion Cracking in the AE44 Magnesium Casting Alloy Using Quantitative Fractography Methods

**DOI:** 10.3390/ma12244125

**Published:** 2019-12-09

**Authors:** Maria Sozańska, Adrian Mościcki, Tomasz Czujko

**Affiliations:** 1Faculty of Materials Engineering and Metallurgy, Silesian University of Technology, Krasińskiego 8, 40-019 Katowice, Poland; 2BGH Polska Sp. z o.o., ul. Żelazna 9, 40-851 Katowice, Poland; adrian.moscicki@bgh.pl; 3Faculty of Advanced Technology and Chemistry, Military University of Technology, Gen. S. Kaliskiego 2, 00-908 Warsaw, Poland

**Keywords:** magnesium alloy, stress corrosion cracking, hydrogen, fractography methods

## Abstract

In this work an assessment of the susceptibility of the AE44 magnesium alloy to stress corrosion cracking in a 0.1M Na_2_SO_4_ environment is presented. The basic assumed criterion for assessing the alloy behavior under complex mechanical and corrosive loads is deterioration in mechanical properties (elongation, reduction in area, tensile strength and time to failure). The AE44 magnesium alloy was subjected to the slow strain rate test (SSR) in air and in a corrosive environment under open circuit potential (OCP) conditions. In each variant, the content of hydrogen in the alloy was determined. The obtained fractures were subjected to a quantitative evaluation by original fractography methods. It was found that under stress corrosion cracking (SCC) conditions and in the presence of hydrogen the mechanical properties of AE44 deteriorated. The change in the mechanical properties under SCC conditions in a corrosive environment was accompanied by the presence of numerous cracks, both on fracture surfaces and in the alloy microstructure. The developed method for the quantitative evaluation of cracks on the fracture surface turned out to be a more sensitive method, enabling the assessment of the susceptibility of AE44 under complex mechanical and corrosive loads in comparison with deterioration in mechanical properties. Mechanical tests showed a decrease in properties after SSRT tests in corrosive environments (UTS ≈ 153 MPa, ε = 11.2%, Z = 4.0%) compared to the properties after air tests (UTS ≈ 166 MPa, ε = 11.9%, Z = 7.8%) but it was not as visible as the results of quantitative assessment of cracks at fractures (number of cracks, length of cracks): after tests in corrosive environment (900; 21.3 μm), after tests in air (141; 34.5 μm). These results indicate that the proposed new proprietary test methodology can be used to quantify the SSC phenomenon in cases of slight changes in mechanical properties after SSRT tests in a corrosive environment in relation to the test results in air.

## 1. Introduction

Magnesium-based casting alloys, with their low density and good technological (workability, castability, machinability, and full recyclability) and mechanical properties (considerable specific rigidity and specific strength), are a very attractive construction material in those industries where it is important to reduce the mass of a structure and at the same time retain its mechanical properties [1,2]. The main examples are the motor and aerospace industries, where magnesium alloys are used for manufacturing various components, such as engine parts, transmission cases, pumps, oil sumps, and steering wheels [3,4,5,6]. Over the last 20 years, there has been a steady tendency to use increasingly strong magnesium alloys, often working at elevated temperatures (above 150 °C) and in a corrosive environment. Such properties of magnesium alloys could be achieved by reducing the content of aluminum (in alloys from the Mg–Al group, such as AZ91, AM50, and AM60) and introducing alloying agents such as Ca, Si, Sr, and rare earth (RE) elements. Thermodynamically stable Al-RE intermetallic phases that form at grain boundaries during solidification (Al_4_RE or Al_11_(RE)_3_ and Al_2_RE and Mg_17_Al_12_) enhanced the mechanical properties (in particular creep resistance). However, their effect on corrosion resistance under SCC conditions in hydrogen-containing environment has not been described.

Literature data [7,8] concerning this problem suggest that the phenomenon may be caused by stresses that are less than 50% of the yield point of particular magnesium alloys being in service, in environment causing only negligible corrosion in those alloys [8,9,10]. Hydrogen-assisted stress corrosion cracking is particularly dangerous for the group of high-strength magnesium alloys containing rare earth elements. Literature data concerning hydrogen degradation and SCC are very scarce for high-strength Mg–Al–RE alloys. This concerns in particular the AE44 casting alloy.

The authors of many publications [1,11,12,13,14,15] agree that the development of stress corrosion cracking in magnesium alloys can take place according to two major destruction mechanisms, which are determined both by the alloy’s microstructure and its operating environment:-continuous crack propagation caused by anodic dissolution of the alloy at the top of cracks,-discontinuous crack propagation caused by a series of cracks occurring at the crack tip.

In the case of the mechanism associated with the continuous dissolution of the alloy at the apex of the crack, several models describing this mechanism can be found in the literature. The most common applies to magnesium alloys characterized by a continuous or almost continuous mesh of phase separation occurring at grain boundaries. The model assumes that material destruction is carried out by galvanic corrosion leading to inter-crystalline stress corrosion cracking [15,16].

The potential difference between the α-Mg solution, which is the anode in this system, and the non-metallic phase precipitates located at the alloy grains, which are the cathode, causes the formation of galvanic cells, which consequently leads to anodic dissolution of the solid solution adjacent to the phase separations. Stresses affecting the material cause continuous tearing of the crack thus created, providing the corrosive environment with constant access to the metallic substrate at the tip of the crack, enabling it to spread continuously.

Another model assumes that in alloys where there is no continuous or almost continuous mesh of phase separation at grain boundaries, a similar process may also occur that causes anodic dissolution of the α-Mg solid solution. However, unlike the model described earlier, it assumes that a galvanic cell can be formed between the passive layer formed on the surface of a magnesium alloy and the place where this layer has been damaged as a result of mechanical stress or an aggressive environment revealing a metallic substrate [16]. In this arrangement, the exposed metallic substrate acts as an anode, while the passive layer is a cathode. As a result of the corrosive cell, a pitting is formed which is also a stress concentrator. Their presence at the crack tip prevents reproduction passive layer, which leads to crack growth. It should be noted that the operation of this model is possible only for alloys and environments in which the repassivation speed is insufficient to restore the protective layer on the alloy surface.

The mechanism of discontinuous crack propagation caused by a series of microcracks at the tip of the crack is quite different. The result of this mechanism is the formation of both trans-crystalline and inter-crystalline stress corrosion cracking. The authors of numerous studies [1,11,12,13,14,15] agree that the impact of hydrogen on the alloy plays a key role in the functioning of this type of destruction, while it is difficult to precisely determine its role. Studies devoted to the phenomenon of stress corrosion cracking suggest that in the case of magnesium alloys, material cracking may occur according to the following models taking into account hydrogen as the driving force of the cracking process [1]:-local decohesion of the crystal lattice caused by hydrogen (hydrogen enhanced decohesion—HEDE),-local hydrogen-induced plasticization (hydrogen enhanced localized plasticity—HELP),-dislocation emission caused by adsorption (adsorption-induced dislocation-emission—AIDE),-delayed hydride cracking (DHC).

Material destruction according to HEDE, HELP, DHC, and AIDE models also takes place in the case of hydrogen embrittlement phenomenon of magnesium alloys. The difference that separates the phenomenon of hydrogen embrittlement and stress cracking is the presence of external mechanical stress in the case of stress corrosion cracking, while the main mechanism of destruction is the interaction of hydrogen deformation [17].

Stress corrosion cracking processes can be analyzed using various criteria and testing techniques, including mechanical tests, electrochemical methods and examinations of the microstructure and fracture surfaces. The most frequently used test is the slow strain rate test (SSRT) [10,18]. Mechanical and corrosive tests related to stress corrosion cracking are often complemented by fractographic examination—a qualitative analysis of fracture surfaces or profiles, including the determination of the nature of the fracture, most frequently using scanning electron microscopy (SEM) methods [19].

In this study, it is attempted to analyze stress corrosion cracking in a modern magnesium-based alloy containing rare earth elements—AE44. The RE mixture in the AE44 alloy contains, apart from Al and Si, also La, Ce and Nd. It is commonly believed that precipitates enriched with rare earth metals may enhance not only creep resistance but also stress corrosion cracking resistance. The susceptibility of AE44 to stress corrosion cracking was assessed by determining changes in its mechanical properties under SSRT tests [5] in a 0.1 M Na_2_SO_4_ solution. The mechanical tests were complemented by an analysis of the alloy’s microstructure and by a quantitative evaluation of changes in the morphology of the fracture surfaces being the result of stress corrosion cracking by means of quantitative fractography methods. The developed method for the quantitative evaluation of cracks on fracture surfaces turned out to be a more sensitive method enabling the assessment of the susceptibility of AE44 under complex mechanical and corrosive loads in comparison with mechanical property degradation.

## 2. Materials and Methods 

The AE44 magnesium-based alloy containing rare earth elements (in wt.%: 4.2 Al, 0.2 Mn, 0.1 Si, 4.2 RE (La, Ce and Nd), balance—Mg) was manufactured by die casting in the form of a 30 mm long rods with a diameter of 12 mm. The microstructure of raw material was analyzed using a Hitachi S-3400N (Hitachi, Tokyo, Japan) scanning electron microscope. All samples before structural examinations were subjected to a following metallographic preparation: -placing the cut samples in a conductive resin using a press;-grinding of sample surfaces on abrasive papers with SiC particles with gradation of grains: 120, 320, 600, and 1200;-rinsing of the ground surface of the samples under running water and polishing with the use of diamond suspensions with the following particle sizes: 6 μm, 3 μm, and 1 μm;-the last stage of sample preparation was polishing in a Al_2_O_3_ suspension with a grain size of 0.05 μm.

Due to the high sensitivity of magnesium alloys to corrosion in water environments, special care was taken to ensure that the time between polishing and drying of samples was as short as possible and not exceed 10 seconds.

The slow strain rate test (SSR) was carried out in accordance with ASTM G129-00 (2013) “Standard Practice for Slow Strain Rate Testing to Evaluate the Susceptibility of Metallic Materials to Environmentally Assisted Cracking” [5]. Cylindrical samples with a length of 20 mm and a diameter of 5 mm have been used. Before the test, samples were cleaned with acetone. The SRT test was performed both in air and in a corrosive environment (0.1M Na_2_SO_4_ solution) under open circuit potential (OCP) conditions, at ambient temperature and at a strain rate of ε = 9∙10^−7^ s^−1^. Following parameters were determined: elongation at failure ε (%), reduction in area at failure Z (%), ultimate tensile strength UTS (MPa) and time to failure t (h).

After SSR test, corrosion products were removed from the fracture surfaces by bathing in a 200 g/L CrO_3_ + 10 g/L AgNO_3_ solution for 1 ÷ 3 minutes (as necessary). The samples were then rinsed three times in acetone using an ultrasonic cleaner. The fracture surfaces were analyzed using a Hitachi S-3400N (Hitachi, Tokyo, Japan) scanning electron microscope. A quantitative characterization of cracks observed on the fracture surfaces was performed using an image analysis program—Met-Ilo^®^ (version 15.3, Instytut Inżynierii Materiałowej, Katowice, Poland) [20]. The analysis comprised an assessment of the morphology and distribution of cracks present on the fracture surfaces and the determination of number and length of cracks.

The quantitative fractography analysis covered entire fracture surfaces. The use of the systemic scanning method (a series of photographs showing the entire fracture surface) and the subsequent arrangement of the images into one high-resolution photograph (Figure 1) enabled the assessment of details on a micrometer scale.

The original authors’ own procedure for the quantitative evaluation of cracks in AE44 comprised:**Stage 1**—the preparation of a high-resolution image of the entire fracture surface, made up of a series of images with marked cracks (cracks had to be marked manually in the GIMP image manipulation program as the greyness differences between cracks and some elements of the structure were insufficient for automatic detection),**Stage 2**—the use of image analysis methods (the correction of brightness and contrast and the creation of a binary image of cracks) to determine the quantitative crack parameters.

Apart from the qualitative and quantitative analyses of the fracture surfaces, a qualitative characterization of fracture profiles was performed. Fracture profiles were obtained by grinding one side of a fracture surface with abrasive paper. Due to the considerable susceptibility of magnesium-based alloys to corrosion in water environments, the time between the polishing and drying of specimens was maximally shortened and did not exceed 10 seconds.

The hydrogen concentration in the specimens was analyzed using a LECO TCHEN 600 (LECO, St. Joseph, MI, USA) elemental analyzer. Hydrogen concentration was analyzed for specimens that were subjected to the SSR tests both in air and in a corrosive environment.

## 3. Results and Discussion

The microstructure of as-cast AE44 is characterized by the presence of fine grains with phase precipitates present mainly at grain boundaries (Figure 2). Moreover, few pores of various sizes are also observed in the structure (Figure 2b). 

SEM observations (Figure 2) revealed typical microstructure of diecast AE44 alloy consisted of fine grains of α-Mg matrix and two sorts of tiny precipitates [21]. The first one was with lamellar morphology (phase type A, Figure 2). The second was coarse with globular shape (phase type B, Figure 2). Micro-zone compositions analysis was performed on these white phases by energy dispersive spectrometer (EDS), and the results show that they contain aluminum and rare earth elements, cerium, lanthanum and neodymium. In our research, it was confirmed, according to [21], that these particles are precipitates of Al_11_RE_3_ and Al_2_RE, respectively.

### 3.1. The Slow Strain Rate (SSR) Test Results

AE44 magnesium alloy was subjected to slow strain rate tests in order to determine its susceptibility to stress corrosion cracking in the applied environment (0.1 M Na_2_SO_4_ solution) and under external mechanical loads. The samples were subjected to strain in air as well as exposure to the corrosive environment under OCP conditions. Such conditions were selected in order to determine the influence of a corrosive environment, corresponding to the working conditions of AE44 alloy, on its mechanical properties [22].

The stress–strain curves obtained as a result of the two SSR test variants for the AE44 samples are shown in Figure 3. They have a normal shape, with an elastic and plastic strain zone, but with no distinct yield point. The highest ultimate tensile strength (UTS ≈ 166 MPa) is recorded for the material subjected to strain in air. 

The samples subjected to strain with simultaneous exposure to the corrosive solution under OCP conditions were characterized by lower strength (UTS ≈ 153 MPa). The best plasticity (ε, Z) was displayed by the samples tested in air (Figure 3, Table 1). The elongation recorded for these samples was ε = 11.9% and the reduction in area was Z = 7.8%. For samples subjected to strain in a corrosive environment under OCP conditions, a slightly lower elongation (ε = 11.2%) and a noticeably smaller reduction in area (Z = 4.0%) were recorded (Figure 3, Table 1).

It can be concluded that the mechanical properties of the AE44 alloy after the test in the air were close to the results obtained by Rzychoń and Kiełbus [23]: UTS – 146 MPa, ε – 7.1%. It should be taken into account that the magnesium alloy AE44 is a casting alloy and its properties strongly depend on the volume fraction of gas pores. Literature data indicate that the greatest deterioration of the mechanical properties of magnesium alloys is observed during their deformation in a corrosive environment, under cathodic polarization conditions, as well as in open potential conditions. This effect is observed for many magnesium alloys (AZ31, Mg-8.6Al, ZK60, AZ91D, and Mg-7%Gd-5%Y-1%Nd-0.5%Zr) in various corrosive environments (sensitivity of Mg alloys in 0.1 M neutral salt solution decreases in the following order: Na_2_SO_4_ > NaNO_3_ > Na_2_CO_3_ > NaCl > CH_3_ COONa) [13,14,24,25,26,27]. 

The results of hydrogen concentration analyses for the AE44 alloy after the SSR tests shows that the hydrogen concentrations for both test variants were comparable (Table 1) and that the hydrogen embrittlement index EIε, determined using Equation (1), was approx. 5.9% for elongation ε and approx. 49% for reduction in area EI_Z_.
EI = │X_H_ − Xair│/Xair × 100%, (1) where: X_H_ is the value of a parameter (e.g., reduction in area Z or elongation ε) in the presence of hydrogen, Xair is the value of a parameter (e.g., reduction in area Z or elongation ε) without hydrogen, i = ε or Z.

The role of hydrogen in the destruction of the AE44 alloy is extremely complex and requires a lot of further research regarding both its penetration process into the alloy, but also subsequent diffusion in the alloy and recombination on the internal surfaces of the pores. In practice, it is possible to have several destruction mechanisms simultaneously with any combination thereof. However, there is no doubt that the hydrogen entering the material as a result of electrochemical corrosion processes is in atomic or proton form. In this form it diffuses in the alloy microstructure, and the microstructure of the alloy and existing structural defects—separation of intermetallic phases, grain boundaries, dislocations and pores—play an important role in the processes of hydrogen diffusion in magnesium alloys. In addition, in rare-earth magnesium alloys, hydrogen can form hydrides with RE elements, and pores also play an important role in magnesium foundry alloys. In the pores, hydrogen may recombine to molecular form, accumulate and cause increased stress in these areas [28]. 

When analyzing the mechanical properties of the AE44 magnesium-based casting alloy, it is necessary to take into account the relationship between the porosity of AE44 alloy and its mechanical properties. Lee et al. [29] demonstrated that the mechanical properties of this alloy greatly depended on the number of pores present in the microstructure. Moreover, the results that they obtained indicated a relationship between the number of pores present on the fracture surface and the plastic and strength properties. The distribution of hydrogen in AE44 material is also affected by porosity. In their work [30], Volkova and Morozova analyzed the chemical composition of gases present in magnesium-based casting alloys containing rare earth elements, trapped at the stage of their manufacture. Their research revealed that hydrogen accounted for as much as 93–95% of the gases’ volume. Bearing in mind that pores present in AE44 alloy may contain hydrogen in the form of molecules, trapped in there during the alloy crystallization, the assessment of hydrogen concentration in the AE44 samples seems to involve a potential for major errors. The presence of hydrogen in pores following the crystallization process in AE44 alloy may also be the reason why there is hardly any difference in hydrogen concentration for the specimens after the SSR tests in air (99.7 ppm) and in the corrosive environment (98.7 ppm) under OCP conditions.

### 3.2. Qualitative and Quantitative Evaluation of the Fracture Surface after the SSR Tests Using The Oryginaly Developed Quantitative Fractography Method

The analysis of the AE44 sample surfaces and fracture surfaces after the SSR tests in air and in the corrosive environment showed major differences. There were a few cracks on the side surfaces of the AE44 sample subjected to strain in air (Figure 4a,b). Their propagation direction was perpendicular to the force applied during the strain test.

On the main fracture surface (Figure 4c,d), there are numerous pores that formed during the casting process, of various sizes reaching even several hundred micrometers (Figure 4c). 

In the case of the specimen subjected to strain in the corrosive environment under OCP conditions, there are cracks and very numerous but shallow corrosion pits on its surface side (Figure 5a,b). On the main fracture surface (Figure 5c,d), there are pores of various sizes (Figure 5c) and numerous cracks (Figure 5d). However, there are also areas free of cracks. The morphology of the fracture surface of the AE44 alloy after the air test is typical—a ductile fracture with numerous pores, the elongated shrinkage porosity and spherical gas (air) porosity. It is comparable with the results of other authors [24]. 

A detailed analysis of the fracture surface of the AE44 specimens subjected to strain in air revealed that the fracture was intercrystalline. There were also many intermetallic phase precipitates on the fracture surfaces (Figure 6a), which were located along grain boundaries in AE44. Cracking along grain boundaries was also observed on the fracture profiles (Figure 6a,b). The analysis of the fracture surfaces of the AE44 specimens subjected to strain in the corrosive solution under OCP conditions revealed a significantly larger number of cracks. Moreover, areas of a transcrystalline nature were observed on the specimen surfaces (Figure 6c,d). According to Atrens et al. [31] fracture surface after SSC of Mg alloy is intergranular (IG-SSC) or transgranular (TG-SSC). A continuous or nearly continuous second phase, typically along grain boundaries, causes IG-SSC by microgalvanic corrosion of the adjacent Mg-matrix. IG-SSC is expected in all such alloys. TG-SSC is most likely caused by interaction of hydrogen with the microstructure. Hydrogen strongly interacts with rare earth elements (RE) in magnesium alloy AE44 to form hydrides with them. It is commonly recognized that transgranular SCC (TGSCC) of Mg alloys is a type of HE. HE models involve hydrogen-enhanced decohesion, hydrogen-enhanced localized plasticity, adsorption-induced dislocation emission, and delayed hydride cracking. The presence of hydride (MgH_2_) phase on the fracture surface designates the DHC mechanism for a friction-stir welded magnesium alloy AZ31 [32].

Cracks were also identified in the alloy matrix (α-Mg), where very often they propagated on parallel crystallographic planes (Figure 7). At the same time, it was observed that in the corrosive environment, the cracking occurred not only through the growth of the main crack, but also through the growth of secondary cracks, which together formed more complex crack configurations (Figure 7a). 

Analysis of the places of occurrence of cracks on fracture surfaces indicates that in the AE44 alloy these cracks occurred in the areas of α-Mg solid solution after SSRT tests in corrosive environment and they were not accompanied by the separation of intermetallic phases (Figure 6). No such cracks were found after testing SSRT in the air. At the same time, it should be remembered that in the presence of a corrosive environment, the nature of the fracture changed from inter-crystalline after SSRT tests in the air to trans-crystalline after tests in a corrosive environment. This fact and the presence of numerous cracks in the alloy matrix after SSRT tests in a corrosive environment may indicate the presence of hydrogen, which caused the matrix brittleness in this place.

On fracture profiles, porosity is observed in the entire volume of the specimens, with the pores being located mainly near the specimen axis (Figure 8). This is typical behavior for pores forming during the casting process.

The presence of local cracks propagating from the side surface deeper into the specimen was detected on the side surfaces of all the AE44 specimens (Figure 9b,d). 

There are many research techniques allow quantitative assessment of surface fractures [18]. However, each of these techniques has its limitations. The origins of quantitative fractography can be found in the works Undervood [19], Wrigth and Karlsson [33], Coster and Chermant [34], and Undervood and Banerji [35]. The authors discuss in some detail the possibilities and limitations in the methods of quantitative description of surface fracture in these works. Generally, there are two groups of quantitative fractography techniques using: -methods based on metallographic cross-sections of the fracture surface, i.e., profilometric analysis,-methods based on images of the fracture surface, i.e., quantitative description of the fracture surface.

The authors of this publication have used both techniques—quantitative description of the profile fracture [7,8] or a fracture surface [9,10,36]. However, the most important thing, regardless of the chosen research technique, is the selection of detail of surface fracture for the quantitative description, e.g., a fraction of the ductile fracture, cracks. 

The basis for the quantitative evaluation of cracks on the AE44 fracture surfaces were binary images of total fracture surfaces with marked cracks (Figure 10). 

The results of the quantitative evaluation of cracks on the fracture surfaces (Figure 10, Table 2) show major differences in the numbers and lengths of cracks depending on the SSR test conditions—in air and in the corrosive solution under OCP conditions.

The fracture surface of the specimen subjected to strain in air is characterized by the smallest number of cracks (141). Their distribution was uneven – the cracks formed small clusters (Figure 10a). Moreover, the cracks are characterized by the greatest average length (34.5 µm). The most numerous group of cracks present on the surfaces of the specimens tested in air were cracks that were 20–30 µm long—they accounted for 37.6% of all cracks. No cracks shorter than 14 µm were found. The longest crack found in these specimens was 114 µm long. Detailed information concerning the crack length distribution on the fracture surface of the AE44 specimen subjected to strain in air is shown in the form of a histogram (Figure 11a). The quantitative analysis of the cracks present on the fracture surface of the specimen tested in the corrosive environment under OCP conditions revealed that they were much more numerous than in the case of the specimen tested in air—there were 900 cracks. They were evenly distributed on the fracture surface (Figure 10b). Their average length was 21.3 µm. The most numerous group of cracks was in the rage of length of 10–20 µm. They accounted for 54.3% of all cracks present on the fracture surface. No shorter cracks were found. The longest crack was 139 µm long and this value was greater than that recorded for the specimens tested in air.

The distribution of cracks on the fracture surface is shown Figure 11b. These results indicate that the presence of the corrosive environment resulted in a greater number of cracks, however, the cracks were shorter and thus their average length was also smaller. An important feature of the method presented in this work is that it covers the entire fracture area (resulting from the final specimen failure) and all cracks (taking into account the image resolution) of very different sizes, and the results obtained can be used to assess the susceptibility of an alloy to the action of a corrosive environment. 

Analysis of the SSC fracture surface in the Na_2_SO_4_ solution demonstrates the active role of hydrogen in the process of initiating and developing microcracks. This can be demonstrated by the presence of an intercrystalline surface fracture (TG-SSC) on SSRT samples in a corrosive environment. The presence of TG-SSC in magnesium alloys (in particular with rare earth elements RE) after the SSRT test in a hydrogenation environment, is considered according to Atrens et al. [31], for the effect of hydrogen interaction with the alloy microstructure. 

Based on the research carried out, it was proposed that the model description of stress corrosion cracking in magnesium alloys AE44 consists of the following stages: -Stage one: as a result of exposure of the alloy to the environment containing hydrogen, due to the difference in its concentration in both media, it diffuses from the environment to the surface layer of the alloy. Hydrogen diffusion is gradually weakened by the Mg(OH)_2_ layer formed on the surface of the alloy, which arises as a result of its exposure to the corrosive environment,-Stage two: as a result of the synergic interaction of the external mechanical loads to which the alloy is subjected, and internal stresses induced by the presence of hydrogen in the material microstructure, cracking occurs in the hydrogen-saturated surface layer of the alloy,-Stage three: the cracks created in the microstructure of the alloy reveal previously covered with Mg(OH)_2_ layer, a metal substrate enabling easier and faster diffusion of hydrogen to the inside of the alloy through the surfaces of freshly formed cracks,-Stage four: the cracks in the material propagate into the material, followed by decohesion of the alloy in the areas of the microstructure located in front of the crack tip, where there is a high concentration of stress caused by external mechanical loads and internal stresses induced by the diffusion of hydrogen. In addition, secondary cracks also develop in these areas.

## 4. Conclusions

In the summary of mechanical test results, obtained under complex mechanical and corrosive loads (SSR) tests in air and in a 0.1M Na_2_SO_4_ corrosive environment and of the qualitative and quantitative evaluation of cracks on the fracture surfaces after the SSR tests, the following conclusions were formulated for AE44 alloy:The impact of mechanical loads and the corrosive environment (OCP) under SSC conditions on the properties of AE44 took the form of deterioration in mechanical properties.The influence of hydrogen on the mechanical properties of AE44 was difficult to evaluate as the measured content of hydrogen in AE44 was the sum of hydrogen that penetrated into the alloy from the solution during the SSR test and hydrogen contained in pores that formed in the alloy during the manufacture process.The qualitative analysis of the fracture surfaces of the AE44 specimens after the SSR tests in air and in the corrosive solution under OCP conditions revealed a significantly greater presence of cracks in the case of the specimens tested in the corrosive environment.The quantitative analysis of the cracks (number, length, and size distribution) in the AE44 specimens after the SSR tests indicated a significant difference between the alloy’s susceptibility to SCC in a corrosive environment and in air. Such clear differences could not be observed when the adopted criterion of the alloy’s susceptibility was the changes in its mechanical properties in SSR tests.The analysis of the results concerning the susceptibility of the AE44 magnesium-based alloy to the action of a corrosive environment under SSC conditions can lead to the conclusion that the results obtained from the quantitative fractography methods are an important complement to the SSR tests (changes in mechanical properties) being the basic criterion for the assessment of an alloy’s susceptibility to the combined action of stresses and a corrosive environment, but also that the developed quantitative fractography method is more unambiguous and more sensitive to changes in the corrosive environment and the mechanical load than the change in mechanical properties during SSR tests.

## Figures and Tables

**Figure 1 materials-12-04125-f001:**
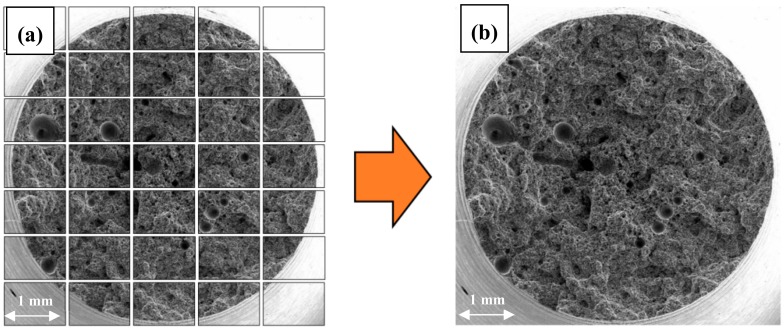
Scheme of the image merging operation (SE) into the image of high resolution (HR) of the fracture surface of the sample using the scanning electron microscope (SEM): (**a**) scheme of systematic scanning operations of the real surface of the fracture - C_ij_ matrix of images, (**b**) image of the total area surface of the fracture after combining images from the C_ij_ matrix.

**Figure 2 materials-12-04125-f002:**
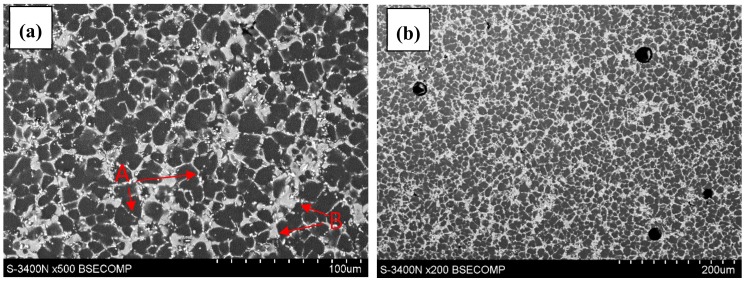
Microstructure of magnesium alloy AE44 in initial state: (**a**) cross-section perpendicular to the axis of the sample; (**b**) casting pores.

**Figure 3 materials-12-04125-f003:**
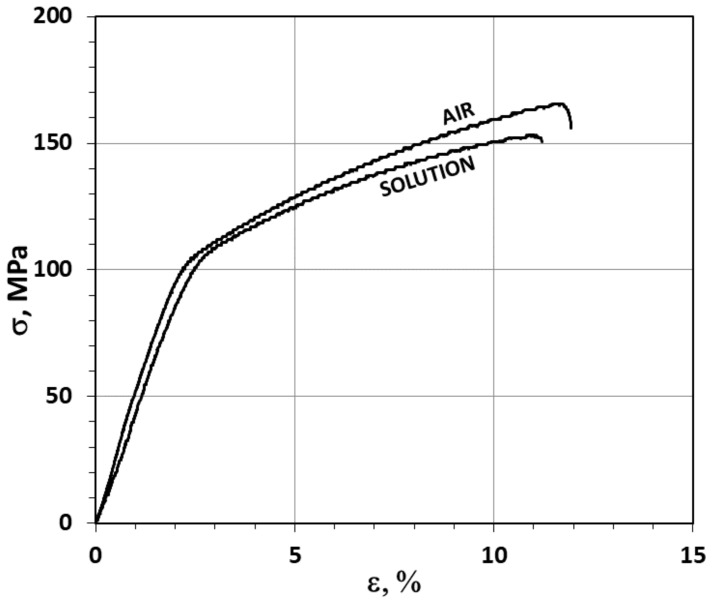
Stress–strain curves (ε = 9 × 10^−7^ s^−1^) for samples of magnesium alloy AE44 deformed in air and in a corrosive environment under open circuit potential (OCP) conditions.

**Figure 4 materials-12-04125-f004:**
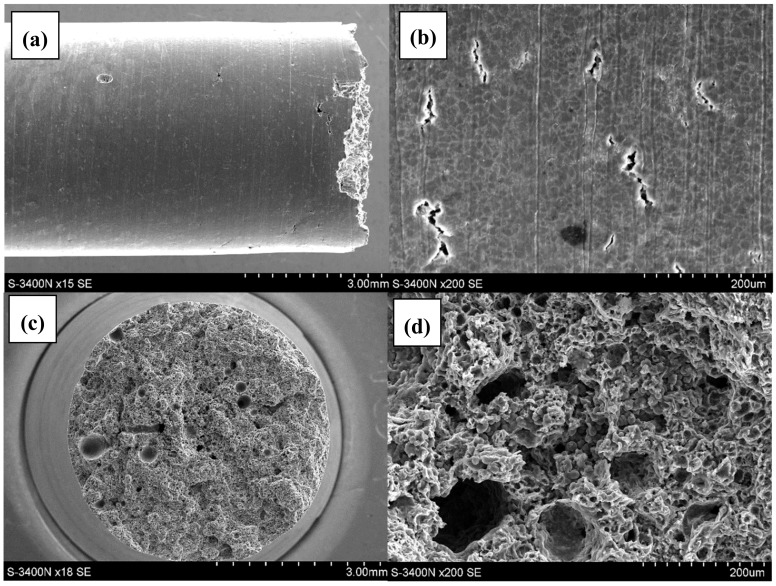
Lateral and fracture surface of the sample after SSR test in air: (**a**) macroscopic image of lateral surface; (**b**) magnified image of the lateral surface; (**c**) macroscopic image of fracture surface; (**d**) magnified image of fracture surface.

**Figure 5 materials-12-04125-f005:**
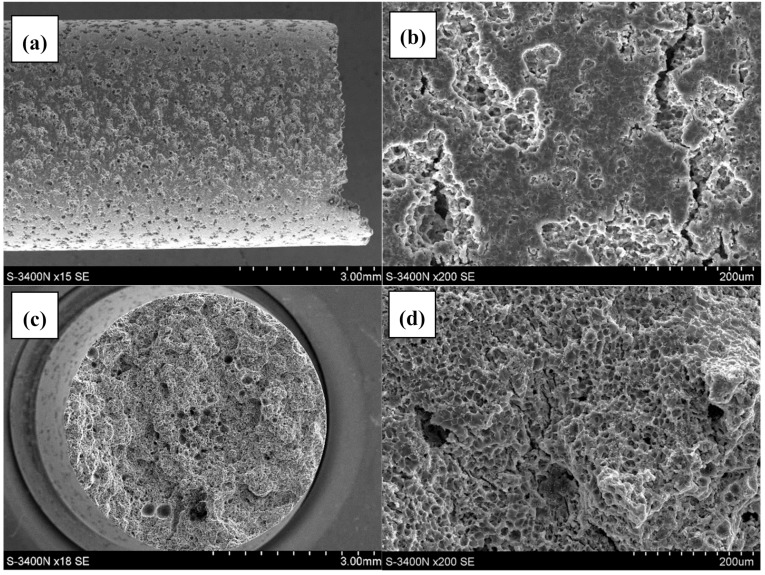
Lateral and fracture surface of the sample after SSR test in corrosive solution under OCP conditions: (**a**)—macroscopic image of lateral surface; (**b**)—magnified image of the lateral surface; (**c**) macroscopic image of fracture surface; (**d**) magnified image of fracture surface.

**Figure 6 materials-12-04125-f006:**
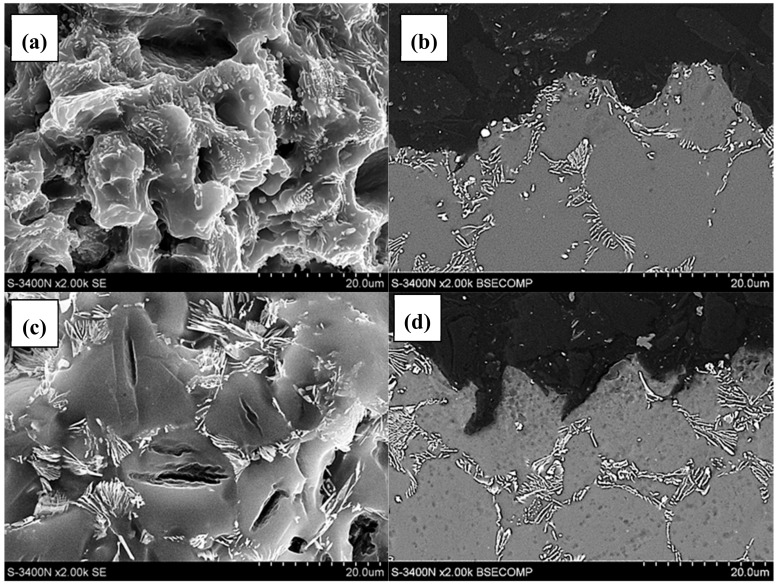
Surface and profiles of AE44 magnesium alloys obtained as a result of the slow strain rate test (SSRT): (**a**,**b**)—air, intergranular fracture; (**c**,**d**)—solution, transgranular fracture, and secondary cracks in the volume of the solid solution α-Mg.

**Figure 7 materials-12-04125-f007:**
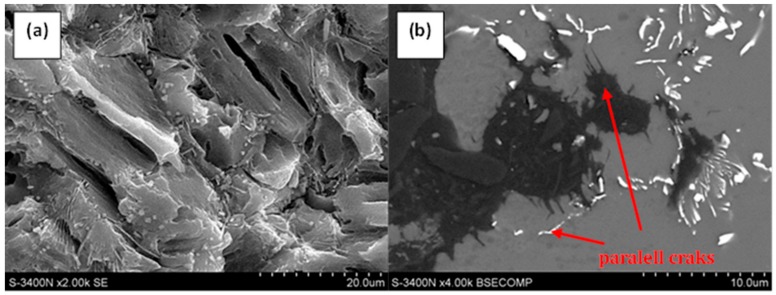
Series of parallel cracks in the AE44 magnesium alloy as a result of SSRT in a corrosive environment under OCP conditions: (**a**) on the fracture surface; (**b**) on the profile of fracture.

**Figure 8 materials-12-04125-f008:**
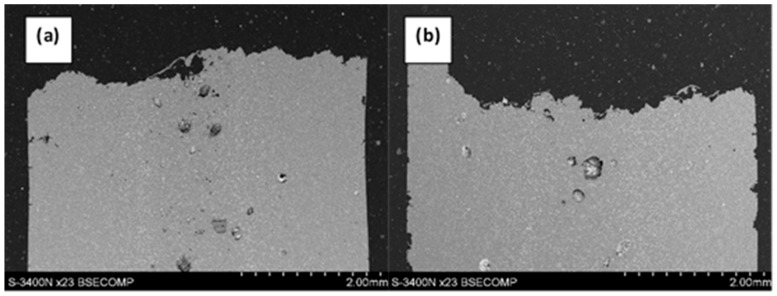
Profile of the fracture in magnesium alloy AE44 after SSRT: (**a**) in the air; (**b**) in solution.

**Figure 9 materials-12-04125-f009:**
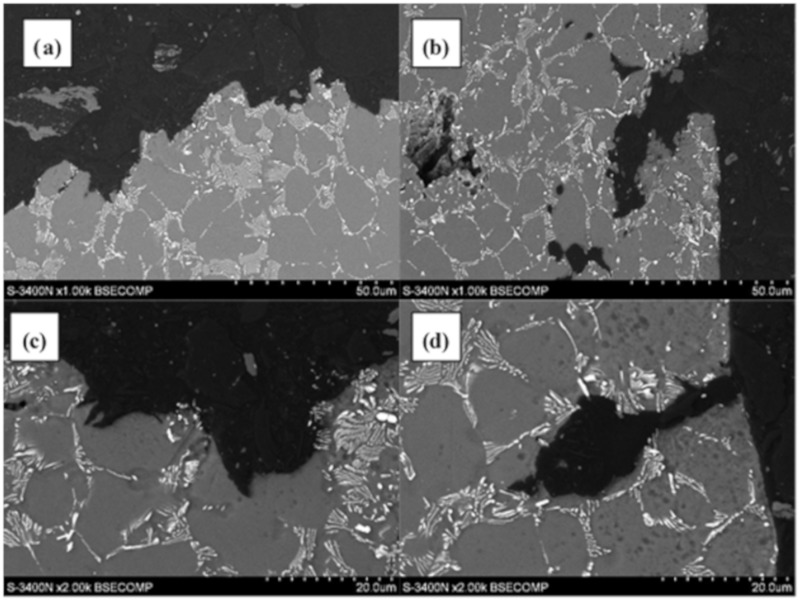
Profile of the fracture in AE44 magnesium alloy after SSRT in the air: (**a**)—fracture surface; (**b**)—lateral surface of the sample and after SSRT in solution: (**c**)—fracture surface; (**d**)—lateral surface of the sample.

**Figure 10 materials-12-04125-f010:**
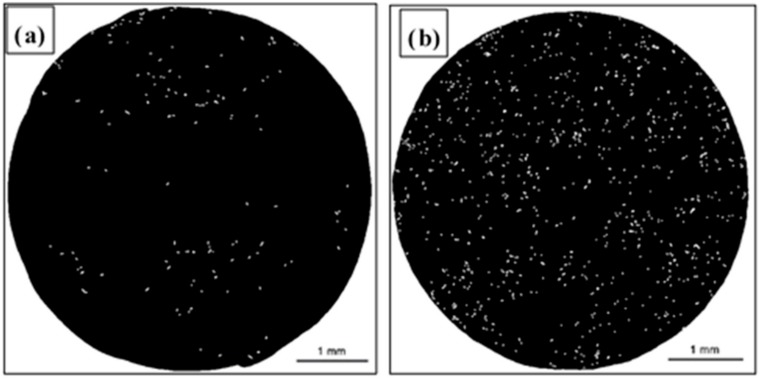
Binary image of cracks on the fracture surfaces of AE44 magnesium alloy after SSRT tests: (**a**) in the air and (**b**) in solution.

**Figure 11 materials-12-04125-f011:**
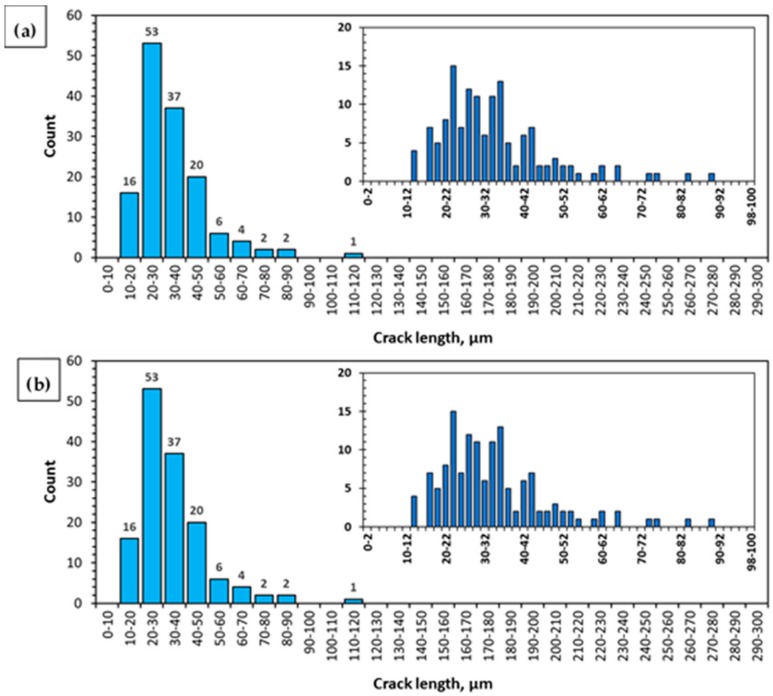
Histogram of the length of cracks occurring in the binary image of cracks present on the fracture surface of the AE44 magnesium alloy after SSRT in the air (**a**) and in solution (**b**).

**Table 1 materials-12-04125-t001:** Mechanical properties, hydrogen concentration C_H_ and hydrogen embrittlement index EI_ε_ for AE44 magnesium alloy AE44 obtained as a result of slow strain rate (SSR) test (ε = 9 × 10^–7^ s^–1^) in air and in a corrosive solution.

Test Environment	Elongation ε [%]	Necking Z [%]	Ultimate Tensile Strength UTS [MPa]	Fracture Time [h]	Hydrogen Concentration C_H_ [ppm]	Hydrogen Embrittlement Index	
						EI_ε_ [%]	EI_Z_ [%]
Strained in air	11.9	7.8	166	39.5	99.7	-	-
Strained in solution	11.2	4.0	153	37.7	98.7	5.9	49

**Table 2 materials-12-04125-t002:** Results of the quantitative analysis of cracks on the surfaces of AE44 magnesium alloy fracture after SSRT tests in air and in a corrosive solution in OCP conditions.

SSRT Environment	Number of Cracks	Length of Cracks [µm]	Width of Cracks [µm]	Distance between Adjacent Cracks [µm]
strained in air	141	34.5 ± 43.8%	1.02 ± 14.1%	118 ± 99%
strained in solution	900	21.3 ± 48.7%	1.04 ± 18.9%	58 ± 76%

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
