# Peer review of "The Characterization of Stress Corrosion Cracking in the AE44 Magnesium Casting Alloy Using Quantitative Fractography Methods"

_materials, 2019, doi:10.3390/ma12244125_

Round 1
Reviewer 1 Report
This paper reports on the effect of a corrosive environment on the mechanical properties of Mg alloys. The reported results are interesting, and the experimental approach is good. However, the main drawback of the paper is that the electrochemical aspects explaining the wet-corrosion process accounting for hydrogen evolution are not well discussed.
More specifically, the authors write about hydrogen but they do not explicitly state if they refer to H2 to H atoms formed as a consequence of the first step of the more complex multistep hydrogen evolution reaction. They also mention the formation of Mg(OH)2 as a consequence of magnesium oxidation but they also have to take into account that such process can passivated areas of the samples where hydrogen evolution is less favoured (see Curioni et al. Electrochimica Acta 274 (2018) 343).
Finally, it is also interesting to know if the distribution of the intermetallic particles is linked to the cracks distribution and if they ca exclude a local embrittlement due to Hydrogen diffusion.
Author Response
The answer for the reviewer comments is submitted in the attachment.

Reviewer 2 Report
The present manuscript entitled “The Characterization of Stress Corrosion Cracking in The AE44 Magnesium Casting Alloy Using Quantitative Fractography Methods” falls under the scope area of Materials journal. My following suggest can help to authors to improve the quality of manuscript.
Abstract needs to rewrite. It does not give any fruitful information about the present study and authors need to provide some quantitative results. Authors need to describe the state of art in introduction section about the corrosion of Mg alloy under SCC in hydrogen environment. Introduction need to elaborate. Is there any specific element in RE of present study? Authors have mentioned that 4.2 RE only. It is not clear that which RE element taken to peruse this work. Authors need to describe about the casting and fabrication of Mg alloy. It is suggested to provide the EDS results of Fig 2 to proof the presence of said elements. It is not enough to mention that it contains RE. It must be well proof.
Author Response
The answer for the reviewer comments is submitted in the attachment

Reviewer 3 Report
Manuscript ID: materials-636338
The Characterization of Stress Corrosion Cracking in the AE44 Magnesium Casting Alloy Using Quantitative Fractography Methods
The manuscript by Sozańska et al. investigates the susceptibility of the AE44 magnesium alloy to stress corrosion cracking in a 0.1M Na2SO4 environment. The manuscript is well presented and authors used well-described methods to investigate the stress corrosion cracking in AE44 Magnesium Casting Alloy. However, there are some points should be addressed before its acceptation for publication in Materials. The following points should be considered:
Authors should explain the difference between the present manuscript and the work published in “Ochrona Przed Korozja, Volume 62, Issue 3, 2019, Page 126”. Transmission Electron Microscopy (TEM) analysis can be done for surface integrity characterization. Nano-hardness measurements can also be performed. Why authors do not perform potentiodynamic polarization tests for corrosion behaviour evaluation.
Author Response

(The authors gave the same response as above.)

Round 2
Reviewer 3 Report
The authors made all the necessary arrangements considering all warnings. it is deemed appropriate to publish this article in this way.